# Development and Assessment of a 4D Printing Technique for Space Applications

**Tim Richter and Christina Völlmecke ***

Stability and Failure of Functionally Optimized Structures, Institute of Mechanics, Technische Universität Berlin, Einsteinufer 5, 10587 Berlin, Germany; t.richter@campus.tu-berlin.de
* Correspondence: christina.voellmecke@tu-berlin.de

**Abstract:** Shape memory polymers (SMPs), a class of polymers exhibiting the unique ability to restore deformation induced during the programming process in response to external stimuli, have garnered significant attention. In this study, our objectives were two-fold: to develop an efficient device for programming SMP hinges crafted from polyetheretherketon (PEEK) and to optimize their performance for potential utilization in space applications. Two versions of the programming device were constructed and compared. Through three systematic experiments, we identified optimal programming and recovery conditions for the hinges, revealing the best shape memory effects (SMEs) at a programming temperature of 250 °C. Remarkably, the hinges were able to recover the previously induced deformation up to 100%, maintaining functionality down to a lower temperature limit of 150 °C. Notably, these hinges demonstrated a wide operational range of over 180°, rendering them promising for space applications, as extensively discussed within the manuscript. However, challenges arise due to the high recovery temperature of 150 °C, presenting obstacles in achieving optimal functionality in the demanding conditions of a space environment.

**Keywords:** PEEK 3D printing; shape memory polymers; aerospace materials; shape memory analysis; self-deploying hinges

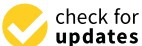



## 1. Introduction

In recent years, 4D printing has emerged as a new fascinating trend within the 3D printing community [1]. This innovative approach to additive manufacturing represents a significant leap forward in the world of material science and engineering. Four-dimensional printing can be best understood as an extension of 3D printing, with the added dimension of time [2]. At its core, it involves the fabrication of objects that have the ability to transform and adapt their shape and properties over time in response to specific stimuli [2,3]. One of the key components of 4D printing in our context is the use of shape memory polymers (SMPs). These polymers exhibit a unique property—they can return to a predetermined shape or configuration when exposed to a particular stimulus, typically heat [4]. This ability to "remember" and revert to a specific shape is at the heart of 4D printing with SMPs.

SMPs, in essence, are materials that have been "programmed" to exhibit this shape memory effect (SME). This programming is a critical aspect of their functionality, and it distinguishes them from traditional materials. It is important to note that the shape memory effect is not an inherent property of the material, but a result of this programming process [1,4]. The allure of 4D printing lies in its potential to design self-deploying objects, offering a wide array of applications. These objects can remain compact and easy to transport before being activated to achieve their intended shape or function [1,4]. This shape altering capability makes 4D printing particularly attractive, as it opens up new avenues for solving complex engineering challenges.

In the medical field, 4D printing has shown promise in the creation of self-assembling medical devices. Yakacki et al. [4] demonstrated the potential of SMPs in the field of minimal invasive surgery in cardiovascular applications.

In the context of space technology, 4D printing takes on a new dimension. The ability to design self-activating and self-deploying mechanisms becomes key. This paper primarily focuses on the application of 4D printing with SMPs in the development of self-activating hinges for space deployment.

While shape memory effects have been utilized in spacecrafts in the past, these applications predominantly involved shape memory alloys (SMAs). SMAs are a class of metals known for their shape memory properties (see [5] and [6] p. 609 ff). In this context, shape memory actuated systems, including hold-down and release mechanisms (HDRMs) and actuators for release and deployment, have been developed using SMAs ([6] pp. 610–614).

It is worth noting that SMPs have gained attention relatively recently in the context of space structures, especially when compared with the more established use of SMAs ([7] p. 47, [8,9]). However, SMPs offer several key advantages over SMAs that make them highly attractive for implementation in space structures.

- SMPs are significantly more cost-effective compared with SMAs [8,10].
- SMPs exhibit a significantly greater recoverable deformation compared with SMAs [10].
- Furthermore, SMPs possess a significantly lower density compared with SMAs, which also enhances their attractiveness for space applications [8,10].

It is worth highlighting that most existing research in the field of SMPs has been centered on their characterization and evaluation using dynamic mechanical analysis (DMA) or dynamic mechanical thermal analysis (DMTA) techniques [4,11,12].

The novelty of this research lies in its distinctive approach to the 4D printing process, which is notable through its seamless execution within a singular device, the A150 from Orion Additive Manufacturing GmbH (https://orion-am.com/, (accessed on 26 September 2023)) (Orion). Unlike conventional methodologies, both the 3D printing of the SMP and its subsequent programming were achieved with this advanced printer. This innovative technique not only simplifies the procedural aspects, but also eliminates the need for an array of expensive equipment for analysis. Traditionally, the characterization of SMEs in polymers demands high-precision equipment, incurring substantial costs. In contrast, the 4D printing process outlined in this study offers a pragmatic alternative, rendering complex analyses more accessible and cost effective. This unique feature positions our research at the forefront of advancements in 4D printing methodologies, setting it apart from conventional approaches in this field.

In response to the identified research gap, the overarching aims of this study are two-fold. Firstly, to develop a programming device designed to facilitate the 4D printing of structures constructed from SMPs, with a specific emphasis on foldable and self-opening configurations. Secondly, to utilize this device for the 4D printing of hinges crafted from PEEK, with the explicit goal of achieving a level of performance suitable for potential deployment in space applications.

To realize these aims, a set of strategic objectives has been defined. The initial objective involves the iterative development of a prototype programming device, focusing on enhancing its design to optimize performance. Subsequently, a crucial step in our methodology is the execution of experiments meticulously designed to evaluate the effectiveness of 4D-printed hinges. These experiments aim to quantify and analyze the performance of the hinges, specifically assessing their ability to maintain and recover specific shapes upon activation. This comprehensive approach to our aims and objectives forms the cornerstone of our investigation, offering a systematic and methodical pathway to contribute significantly to the advancement of space technology. The implications of our research extend beyond the immediate context of spacecraft, demonstrating the versatility and broader engineering significance of self-activating hinges, with potential applications reaching into other domains.

## 2. Materials and Methods

### 2.1. Materials: PEEK

The SMP used in this work is polyetheretherketon (PEEK). PEEK is a semi crystalline polymer [13,14] of a material class, which is often referred to as high performance or engineering polymers [15]. These materials are called engineering polymers as they exhibit exceptional mechanical and often also chemical properties, which allows them to be used in demanding applications that were traditionally reserved for metal, ceramic, or composite materials [16,17]. PEEK, for example, has a relatively high ultimate tensile strength of around 90 MPa [13,15] with a pronounced elastic range and it is resistant to most chemicals and radiation [15–17]. PEEK is gaining more and more attention in lightweight engineering due to its high strength-to-density ratio, which is comparable to that of steel [17]. In this context, PEEK has also already been tested on the International Space Station (ISS) for potential applications in space [18] and was deemed valid for use in spacecrafts due to its high radiation resistance [18].

Historically, PEEK parts were mainly produced through methods like injection molding or CNC milling. However, there is a growing trend towards using 3D printing, specifically fused filament fabrication (FFF), for manufacturing PEEK components. This is significant because PEEK is a high-cost material. FFF's appeal, especially when compared with CNC milling, lies in its material usage approach. FFF precisely deposits the material where needed, as opposed to milling, which subtracts the material. This offers the potential for substantial cost savings when working with PEEK.

Concerning its shape memory characteristics, PEEK has already been researched by scientists because of its heightened mechanical and chemical properties compared with traditional SMPs [17]. The shape memory properties of PEEK can be traced to its semi crystalline nature, where the crystalline phase acts as the permanent network and the amorphous regions act as the thermally activated switching or transition segments [17,19]. The $T_{\text{trans}}$ for PEEK is traditionally the $T_g$, which is around 150 °C. However, it is worth noting that the observed shape memory properties of PEEK have been relatively small [17].

In the scope of this study, PEEK was chosen as the SMP due to its prior testing on ISS and NASA's approval for deployment in spaceflight missions [18]. Given that the primary aims of this research is to create hinges suitable for space environments, the selection of a material capable of addressing the specific challenges inherent in such conditions is crucial. The specific PEEK filament material used in this work for the 4D printing procedure was kindly provided by Evonik Industries AG (www.corporate.evonik.com, (accessed on 20 September 2023)) and is called "Evonik Infinam PEEK 9359 F". This PEEK filament was specifically developed for industrial and aerospace applications; the material characteristics from the data sheet can be seen in Table 1.

**Table 1.** Material characteristics of Evonik Infinam PEEK 9359 F filament, data from the datasheet provided by Evonik [14].

| Description | Value |
|---|---|
| Tensile Modulus | 3600 MPa |
| Yield stress | 90 MPa |
| Yield strain | 5% |
| Melting temperature | 340 °C |
| Glass transition temperature | 152 °C |
| Density | 1300 kg m$^{-3}$ |
| Filament Diameter | 1.75 mm |

Aside from PEEK, PLA was also used in the scope of this work. PLA is a biopolymer that is widely used in the 3D printing scene, especially for manufacturing prototypes.

In this work, the material was only used for preliminary investigations and for form fitting purposes. For further information on the material, refer to [20,21].

### 2.2. 3D and 4D Printing

Three-dimensional printing is a relatively new manufacturing method compared with other manufacturing techniques. In 3D printing material is deposited in layers and lines; multiple of these lines are deposited next to each other, forming layers, which are deposited on top of each other forming the final part geometry. In this work, the primary focus will be on the polymer 3D printing technique known as fused filament fabrication (FFF), also referred to as FDM. FFF is categorized as a material extrusion-based 3D printing method. In FFF, a thermoplastic polymer in filament form is extruded through a nozzle. The filament undergoes a melting process within the nozzle and is then deposited onto a build platform, typically in linear patterns, while in its molten state. The polymer subsequently solidifies, capitalizing on the thermoreversible networks inherent in thermoplastic materials.

The nozzle is an integral component of the 3D printer's print head, which maneuvers along three axes, enabling the production of three-dimensional objects. After the material is deposited onto the build platform, it undergoes cooling. When the new material is deposited adjacent to or above the previously deposited layer or line, the thermal energy from the freshly deposited material facilitates (partial) fusion with the existing material [22].

In the context of this study, 3D printing serves as the pivotal method for fabricating the hinges crucial to our investigation. The hinges, designed as a simple rectangular geometry, were initially conceived as a baseline for subsequent explorations into more suitable hinge shapes. Surprisingly, this basic geometry exhibited an exceptional performance in our preliminary tests and, as a result, became the focus of our experimentation.

The selected hinge geometry, measuring $90\,\text{mm} \times 24\,\text{mm} \times 0.4\,\text{mm}$, was deliberately crafted to be thin and long This specific design, featuring a slender structure, was proven to be advantageous for the intended application as a self-opening hinge. The inherent high deformability of this geometry aligned with the requirements of the intended use case. The unfolded and folded hinge are displayed in Figure 1.

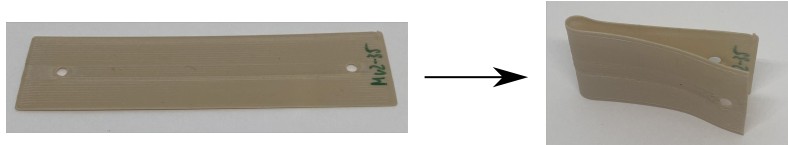

**Figure 1.** The unfolded hinge before programming on the left and the folded programmed hinge on the right.

When working with 3D printers, there are three steps in creating the final 3D printed part.

1. **CAD modelling:**
   The hinge geometry was initially modelled using computer-aided design (CAD) software (https://www.solidworks.com, (accessed on 14 November 2023)). During this phase, careful consideration is given to the orientation in which the part will be printed to avoid challenges such as overhangs or difficult-to-print sections.

2. **Slicing:**
   Once the CAD model was finalized, it was exported to a .stl file, creating a meshed version of the part. This file was then loaded into a slicing program, in this case, Cura. In the slicer, the printing process was planned, and various process parameters were fine-tuned based on the specific part geometry. Common parameters included print speed, print temperature, and line width, which significantly influenced the final printing outcome. The slicing software generated a G-code file containing the instructions for the 3D printer.

3. **3D printing on Orion's A150:**
   The 3D printer employed in this study was developed by Orion Additive Manufacturing GmbH (Orion). Orion's printer surpasses traditional FFF capabilities with its

innovative system specifically designed for engineering polymers like PEEK. Notably, it achieves nearly 100% density and isotropic strength values that are comparable or even better than those of injection-moulded parts [15]. The key to this success lies in Orion's thermal radiation heating (TRH) system, maintaining a warm printing chamber. Active heating during printing ensures the effective fusion of layers, minimizing air pockets. Refer to Figure 2 for an illustration of one of Orion's printers used in manufacturing and testing the hinges.

The G-code that was maintained after slicing was then utilized by the 3D printer. The printer read the file, interpreting each line as instructions for movement or changes in process parameters, thus gradually printing the hinge.

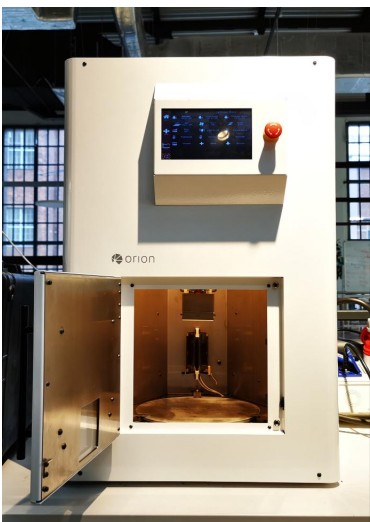

**Figure 2.** Orion's medical grade A150 3D printer, taken from [15]. Outer dimensions: 570 mm × 550 mm × 830 mm. Build volume: 180 mm diameter × 150 mm high.

The hinge geometry, designed to be simple yet effective, coupled with the advanced capabilities of Orion's A150 3D printer, laid the foundation for our experiments on 4D printing of foldable, self-opening structures. The emphasis on simplicity in geometry served as a starting point for exploring the complex interplay between design, material properties, and the 4D printing process. The hinge geometry for the second iteration bending device underwent slight modifications. The holes at the end were strategically positioned to avoid any interference with the bending process. Consequently, the overall geometry did not exert any influence on the bending results. The configuration depicted in Figure 3a corresponds to the bending device illustrated in Figure 4, whereas the configuration in Figure 3b corresponds to the device presented in the results section.

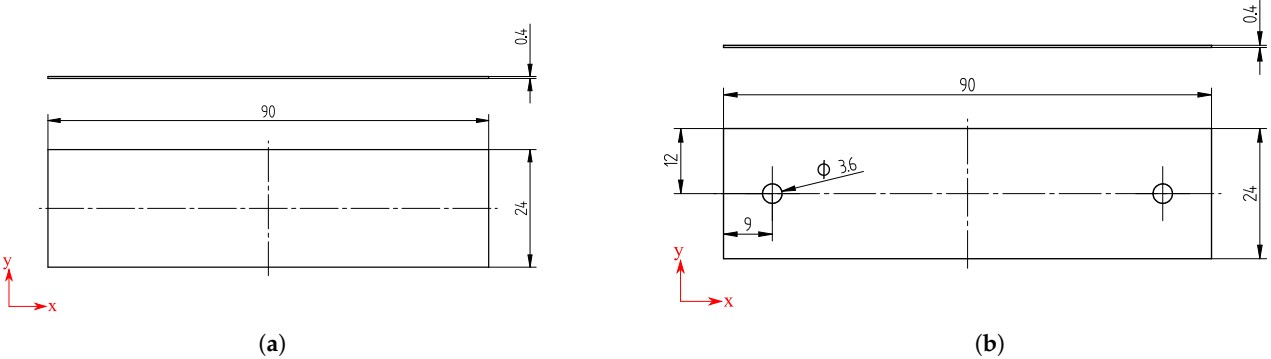

(a)          (b)

**Figure 3.** Technical drawing of the unfolded hinges used for the experiments in millimeters. (**a**) The hinge geometry for the first iteration programming device. (**b**) The adjusted hinge geometry for the second iteration programming device.

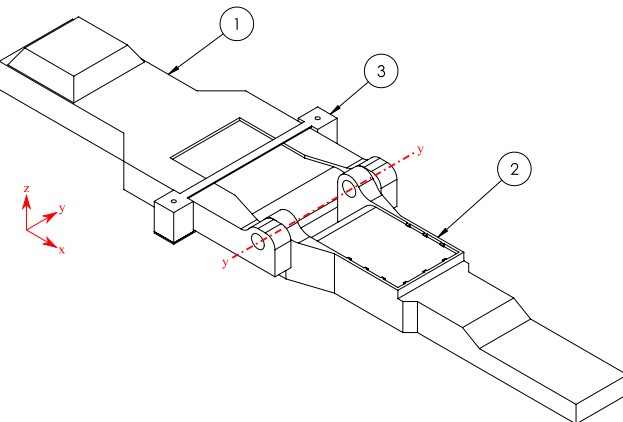

**Figure 4.** Isometric assembly drawing of the programming device **version 1** used in the experiments. With the components: (1) the base lever, (2) the rotating lever, and (3) the slidable hold down mechanism. Outer dimensions: 232 mm × 67 mm × 16 mm.

### 4D printing

The term 4D printing, coined by Skylar Tibbits a decade ago [2], represents a paradigm shift in additive manufacturing. It involves the creation of multi-material prints with the unique ability to transform over time. Tibbits defines it as a process enabling customized material systems to change from one shape to another directly off the print bed [2].

Since its introduction, 4D printing has evolved into a dynamic field of research [1]. The initial definition by Tibbits has undergone several refinements, leading to a broader context. In modern terms, 4D printing includes structures that, post-printing, can autonomously alter their shape in response to specific stimuli [1]. This includes the utilization of SMPs, a crucial aspect of 4D printing.

### Shape Memory Effects in 4D Printing

Shape memory effects (SMEs) play a pivotal role in the functionality of 4D printing, particularly when employing SMPs. SMPs possess the unique ability to recover deformation in response to a stimulus, with thermal energy being the main focus in this context [23]. SMEs in SMPs result from a carefully orchestrated material treatment process, known as a shape memory cycle.

This cycle involves three main steps:

1. **Programming:** The polymer is initially in its undeformed state (shape A) and is subjected to thermal energy above a specific transition temperature ($T_{\text{trans}}$). This allows the polymer to undergo deformation under an applied force. Upon cooling below $T_{\text{trans}}$, the deformed state is fixed [24].

2. **Storage:** The polymer, now in its deformed temporary state (shape B), stabilizes the deformation. This stage allows for convenient storage, with the polymer retaining its altered shape [24].

3. **Recovery:** Reheating the polymer above $T_{\text{trans}}$ reactivates its ability to recover the initial deformation, completing the shape memory cycle [24].

The underlying polymer structure involves macromolecules with specific configurations. Understanding these configurations is essential for harnessing SMEs effectively.

In the realm of heat-induced SMPs, three main groups exist [19]: the dual-state mechanism (DSM), dual-component mechanism (DCM), and partial-transition mechanism (PTM). DSM involves a glassy polymer transitioning between rubbery and glassy states, with examples like PMMA and silicone [19]. DCM employs a dual-component system, combining elastic and transition segments within the polymer [19], while PTM relies on the partial heating of the polymer, which supports/enables the SMEs. In the realm of FDM

*2.3. Design of Experiments*

In this study, our primary aims were two-fold. First, we aimed to develop a device that enables the programming and hence the 4D printing of foldable, self opening structures. Second, we aimed to 4D print hinges with the previously mentioned device using PEEK, with the potential application as self-deploying hinges in space environments. We sought to demonstrate that SMEs in PEEK were sufficient to enable a fully self-deploying system. Notably, prior research ([17]) cast doubt on this hypothesis. Should the SMEs prove insufficient for the desired application, we will explore methods for optimizing the achievable SMEs.

Our experimental design involves the following steps:

1.  **Hinge Fabrication:** Hinges made from PEEK will be 3D printed on Orion's A150 3D printer to ensure high density and isotropy. Because of confidentiality agreements established with Orion, specific parameters employed for 3D printing the hinges on Orion's printers cannot be disclosed at this point. This is integral to protect proprietary details, preserving Orion's competitive edge, and protecting their techniques from unauthorized replication or exploitation.

2.  **Programming Process:** After printing, the hinges will be programmed using the build chamber of Orion's 3D printer. The programming process involves heating the hinges and utilizing a custom-made programming device. This device will be constructed and optimized to efficiently support the programming of hinges.

3.  **Device Iteration:** Following an initial set of experiments, the programming device will be altered and optimized, and all experiments will be repeated using this updated device. This device will also be 3D printed using PEEK to match the high-temperature environment required for programming. The device was printed using a rectangular infill pattern with an infill density of 30%.

4.  **Experiments:** The experiments are designed to assess the shape memory characteristics of the hinges, with a focus on programming and recovery:

    (a)  **Programming Assessment:** The hinges will be programmed at different temperatures inside the print chamber (150 °C, 175 °C, 200 °C, and 250 °C) and for varying time durations (5 min, 10 min, and 20 min). The success of the programming process will be measured using the achieved fixed angle, denoted as $\alpha$. This angle, as illustrated in Figure 5d), represents the angle between the inner two surfaces of the hinge after programming, with 0° being the smallest possible angle (perfect programming) and 180° being the largest possible angle (no programming at all). The fixity ratio ($R_f$) is calculated using the formula:

    $$R_f = \frac{180° - \alpha}{180°} \quad (1)$$

    Subsequently, the hinges will recover their deformation at the same temperature and time duration as the initial programming. The recovered angle $\beta$ will be measured to calculate the recovery ratio ($R_r$). $\beta$ is the angle between the inner two surfaces of the hinge, with 180° being the largest possible angle (perfect recovery) and $\alpha$ being the smallest possible angle (no recovery at all). The recovery ratio ($R_r$) is calculated using the formula:

    $$R_r = \frac{\beta - \alpha}{180° - \alpha} \quad (2)$$

    This experiment aims to determine the optimal temperature and minimum time required for effective programming.

    (b)  **Recovery Assessment:** Once the optimal programming conditions are determined, the hinges will be tested to measure their recovery potential. After successful programming at the identified optimal conditions, the hinges will be subjected to recovery tests at various lower temperatures ($-10$ °C, $-30$ °C,

−50 °C, and 150 °C), with the recovery time measured. The programming and recovery ratios will be calculated. This experiment aims to establish the lower limit for efficient recovery.

(c) **Forced Cool-Down Assessment:** In this experiment, the hinges will be programmed at a temperature that did not yield ideal results for fixity and recovery. Subsequently, the hinges will be rapidly cooled by submersion in water. The results will be compared with a reference group programmed in the conventional manner. The goal of this experiment was two-fold; first, investigate what influence the rapid cool-down has on the performance of the hinges and, second, see if the cool-down time and ease of handling can be improved.

Our experimental design seeks to optimize the number of experiments while ensuring comprehensive data. The first experiment spans a broad temperature range, determining the optimal programming temperature. The second experiment validates the programming process at the ideal temperature, providing ample data and exploring recovery at different temperatures. The third experiment investigates the effectiveness of a rapid cool-down method, with a larger sample size. The process of how the experiments are conducted is also visualized in Figure 5.

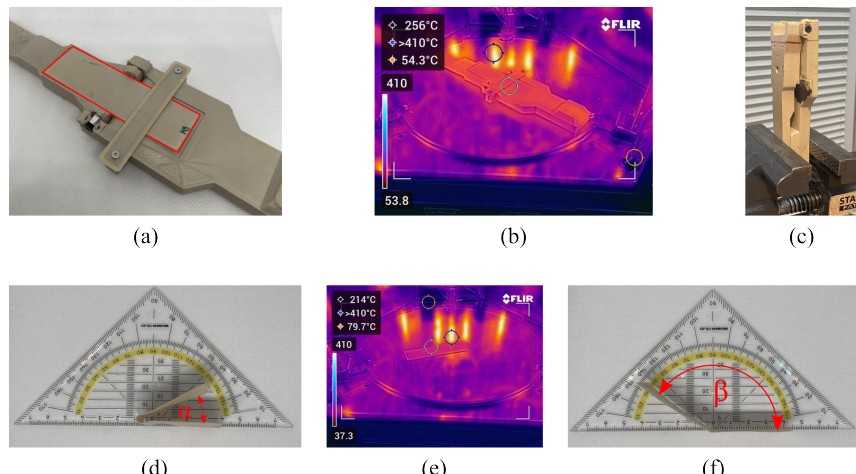

(a)   (b)   (c)

(d)   (e)   (f)

**Figure 5.** The process chain for the experiments: (**a**) placing the hinges (marked in red) in the programming device (both made from PEEK), (**b**) heating the hinges and the device and measuring the surface temperature, (**c**) bending/folding the hinges and letting them cool down in the vice to keep the stress applied (alternatively, in the third experiment on the influence of rapid cool down, the hinges and the device are submerged in water), (**d**) measuring the achieved fixed angle $\alpha$, (**e**) reheating the hinges, and (**f**) measuring the achieved recovered angle $\beta$.

**Apparatus and Setup:**

The following equipment and devices were employed in our experiments:

- Orion's A150 3D printer for 4D printing.
- A simple triangular ruler to measure the achieved programmed and recovered angles.
- A thermal imaging camera from FLIR Systems Inc. (Flir C5) to measure and validate temperatures inside the printer.
- A custom-built programming device used for hinge programming.
- High-temperature gloves for taking the programming device out of the printer.

Our experiments involved the manipulation of several independent variables, including:

- The temperature set for the build chamber.
- The exposure time (the duration hinges were left inside the build chamber).
- The generation of the programming device used.

Dependent variables measured include:

- Programmed angle $\alpha$, used to calculate the fixity ratio ($R_f$).
- Recovered angle $\beta$, used to calculate the recovery ratio ($R_r$).
- Recovery time (specific to Experiment 2).
- Surface temperature measured before programming using the thermal imaging camera.

After 3D printing, the hinges were stored in a Totech Super Dry SDB-1106-40 drying rack to protect them from moisture. For programming, the hinges were securely fixated inside the programming device and subjected to heating within the build chamber.

To minimize the influence of manufacturing batches, hinges were randomized to ensure data integrity. Replication can be easily achieved using the provided STL and CAD files for both the programming device and hinges.

Data analysis was performed using Python, Seaborn, and Scipy. Figures were designed with consideration for readability among individuals with common color blindness conditions, and tested using the Coblis Color Blindness Simulator.

### 3. Results

*3.1. Programming Device*

In the pursuit of enhancing the 4D printing process for SMP hinges, a programming device was meticulously developed through an iterative design approach. The overarching goal was to facilitate the effective programming of SMP hinges for subsequent deployment in space applications.

The design process commenced with preliminary tests utilizing PLA, and the final version of the programming device was materialized using PEEK. This section focuses on two distinct versions of the programming device, elucidating the evolution between them. Later, we explore the impact of the two versions on improving the 4D printing process.

**Shared Features:**

Both iterations of the programming device share fundamental features essential to their functionality.

- **Dual Handles:** Each version incorporates two handles—a larger one referred to as the base handle and a smaller rotating handle. These handles are interconnected through a press fit along the $y$–$y$ axis, achieved by embedding an M3 thread insert into connecting holes. For hinge programming, these two handles are pulled towards each other, allowing the hinge inside to be folded.
- **Vertical Alignment Feature:** Positioned at the end of the base handle, this feature prevents over-tightening when the device is secured in a vice for cooling. Conversely, the rotating handle features a negative counterpart of this alignment feature.

**First Version:**

The initial iteration of the programming device (version 1) adopts a flexible design for hinge fixation, allowing for quick insertion and removal. Key features of the first version include teeth and slidable hold-down mechanism. The rotating handle incorporates small teeth to secure hinges, while the base handle features a slidable hold-down mechanism aiding in $z$-direction stability (refer to Figure 4).

Programming Steps (version 1):

1. Slide back the slidable hold-down mechanism on the base handle.
2. Load the hinge into the programming device, ensuring the teeth grip the hinge.
3. Slide the hold-down mechanism back above the hinge, securing it on the base handle side (shown in Figure 5a).
4. Place the device with the loaded hinge in the build chamber of Orion's A150 and heat (see Figure 5b).
5. Quickly remove the device from the 3D printer.
6. Pull both handles towards each other, effectively folding the hinge, while at the same time sliding back the hold-down mechanism on the base handle side.

7. Clamp the device into a vice (for experiment 3 submerge in water) with the hinge still inside and allow both the device and hinge to cool down (Figure 5c).
8. After cooling, extract the device from the vice, revealing the folded hinge (Figure 5d).

**Second Version:**

The subsequent iteration employs a more static approach, screwing the hinges into the device. Noteworthy features of the second version include a screwed fixation, a non-slidable hold-down mechanism and cooling holes. The rotating handle has an M2 thread insert for hinge fixation using an M2 countersunk screw. The base handle also features a hole for tool access. The non-slidable hold down mechanism is constructed to be very thin and flexible. Once it is screwed in place, it acts as a subtle interference layer between the hinge faces during folding, and keeps the hinges secured in the z direction. Both handles of the second version feature holes to enhance cooling efficiency during the 4D printing process. All of these features are also visible in Figure 6.

Programming Steps (version 2):

1. Carefully slide the hinge under the hold-down mechanism on the base handle.
2. Align the holes in the hinge with those on the programming device.
3. Use an M2 countersunk screw to fixate the hinge in the rotating handle.
4. Place the device with the loaded hinge in the build chamber and heat.
5. Quickly remove the device from the 3D printer.
6. Pull both handles towards each other, effectively folding the hinge.
7. Clamp the device into a vice (for experiment 3 submerge in water) with the hinge still inside, allowing both the device and hinge to cool down.
8. After cooling, extract the screw using an allen key, enabling the device to open and revealing the folded hinge.

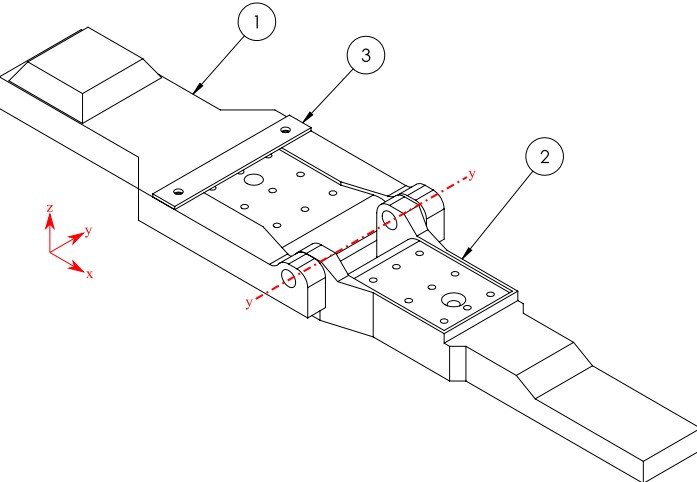

**Figure 6.** Isometric assembly drawing of the programming device **version 2** used in the experiments with the following components: (1) the base lever, (2) the rotating lever, and (3) the fixed hold-down mechanism. Outer dimensions: 232 mm × 50 mm × 16 mm.

This detailed examination of the programming device establishes a foundation for subsequent sections, where the impact of these design variations on the 4D printing process will be scrutinized.

### 3.2. Programming Assessment

The first investigated phenomenon within the first material test was the reaction of the hinges to different exposure times $t_e$. Figure 7 shows, on the $x$ axis, the time the hinges were exposed to the heat inside the print chamber and, on the $y$ axis, the measured surface temperature, using the previously mentioned thermal imaging camera, $T_S$, after being

exposed, to that heat. The different line colors visualize the temperature $T_{set}$ that was entered on the printer's interface.

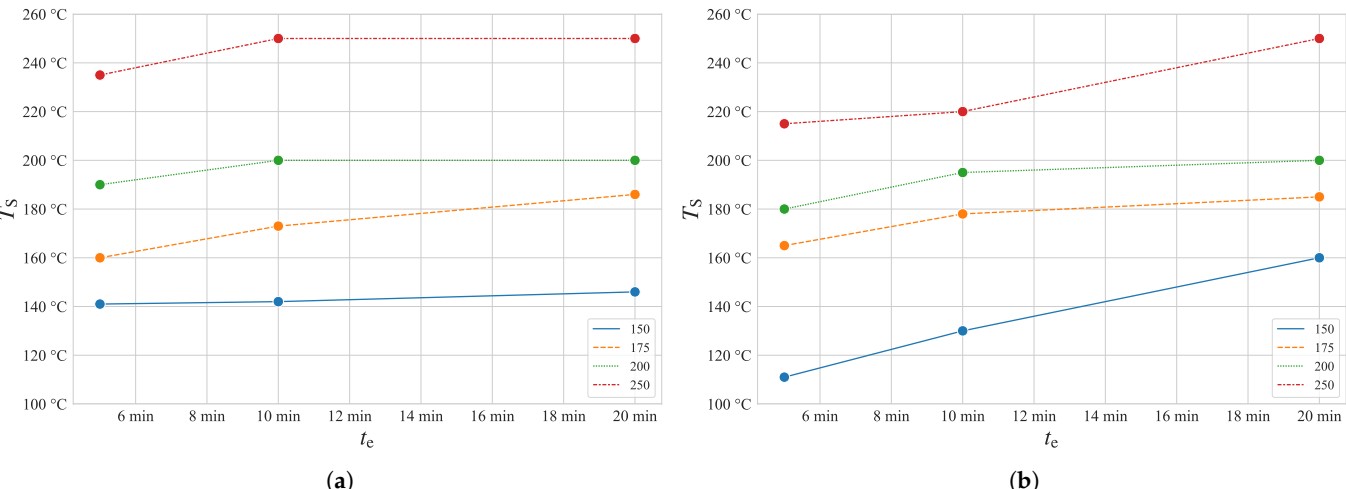

(**a**)                                         (**b**)

**Figure 7.** Measured Surface temperature $T_S$ on $y$-axis, after 5 min, 10 min and 20 min of exposure time ($x$-axis) and different set temperatures on the printer's interface $T_{set}$ (hue); for (**a**) the first iteration of programming device and (**b**) the second generation programming device.

First, we can see that the entered temperature was almost never consistently accomplished, but instead the surface temperature only lay within a range of the entered temperature. Depending on the heating time, the measured surface temperature was sometimes below the entered temperature, but in some cases it was even above the entered temperature.

Next, we investigated how well the hinges could be programmed at their measured surface temperature, for that, Figure 8 provides valuable insight.On the $x$ axis of Figure 8 we can see the recorded surface temperature $T_S$ and on the $y$ axis the fixity ratio calculated using Equation (1) is shown. The two colors show whether a data point belongs to the experiment conducted with the first iteration programming device (blue) or the second iteration (orange). In addition to the raw data points, a linear regression was used to visualize possible trends in the data set, which is displayed as a line in Figure 8. The colored area around the lines represents the 95% confidence interval. Of course, the linear regression line can only sensibly be interpreted in the range of the 0% . . . 100% fixity ratio; therefore the data above 100% were cut off.

What we can clearly see in Figure 8 is that the higher surface temperatures were highly correlative with high fixity ratios. This data also held statistical significance with $p_{v1} = 1.6 \cdot 10^{-5}$ and $p_{v2} = 3.3 \cdot 10^{-5}$.

Similarly, we can also compare the temperature set on the printer's interface with the fixity ratio, see Figure 9.

After investigating the programming behavior, the hinges were also allowed to recover at the same $T_{set}$ and $t_e$. The achieved recovery ratio in comparison with the measured surface temperature before programming is visualized in Figure 10.

The scaling of the $y$ axis in Figure 10 was adjusted compared with the other graphs to better visualize the phenomena at hand. It should be underlined, that the temperature on the $x$ axis does not display the actual recovery temperature, which was not measured, but instead the temperature the hinges were programmed at, thus allowing for relating each data point to the previous graph (Figure 8). However, as the hinges were recovered at the same $T_{set}$ and $t_e$, the surface temperature during recovery could be expected to be similar to the one right before programming.

We can see that, similar to the programming of the hinges, the recovery behavior also seemed to correlate to the temperature. Notably, the recovery behavior of the hinges

was only a bonus observation that was recorded, because the hinges were only used once for programming.

To visualize the connection between the programming and recovery behavior, we plotted the recovery ratio against the fixity ratio, as shown in Figure 11.

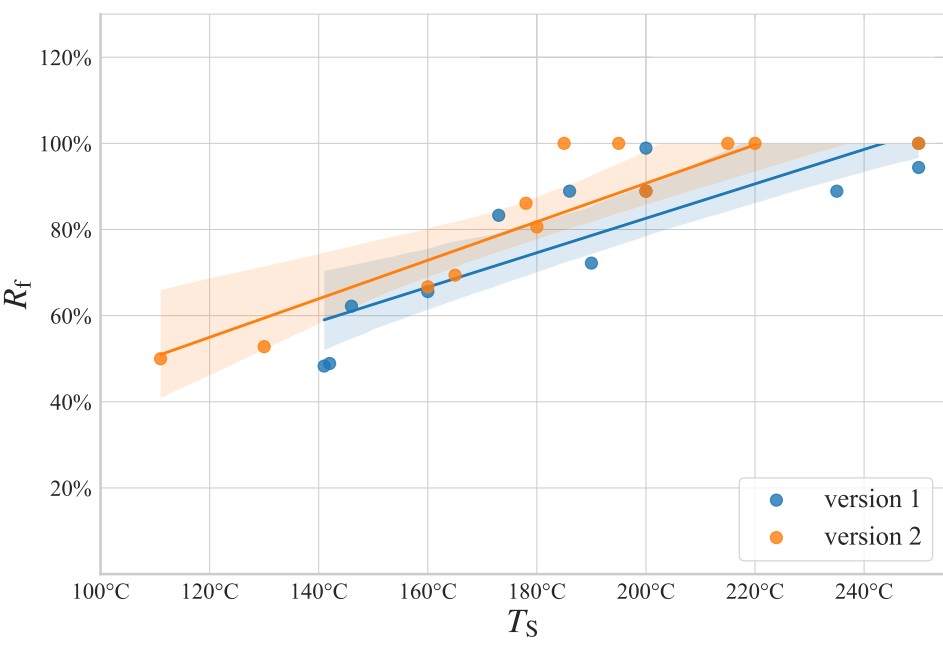

**Figure 8.** The calculated fixity ratio $R_f$ in relation to the measured surface temperature $T_S$ with a linear regression and a 95% confidence interval.

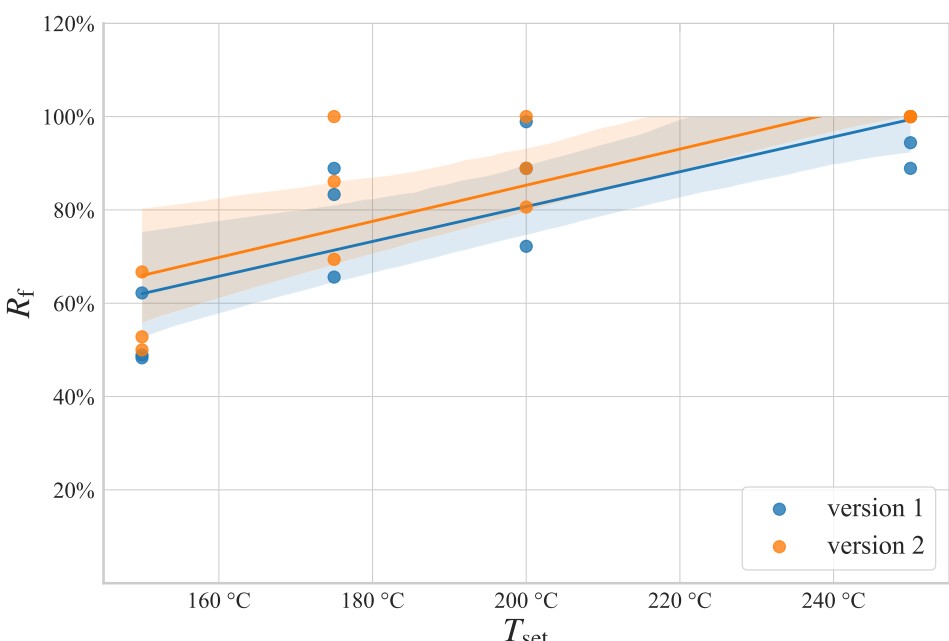

**Figure 9.** The calculated fixity ratio $R_f$ in relation to the temperature set on the printer's interface for the build chamber, $T_{Set}$ with a linear regression and a 95% confidence interval.

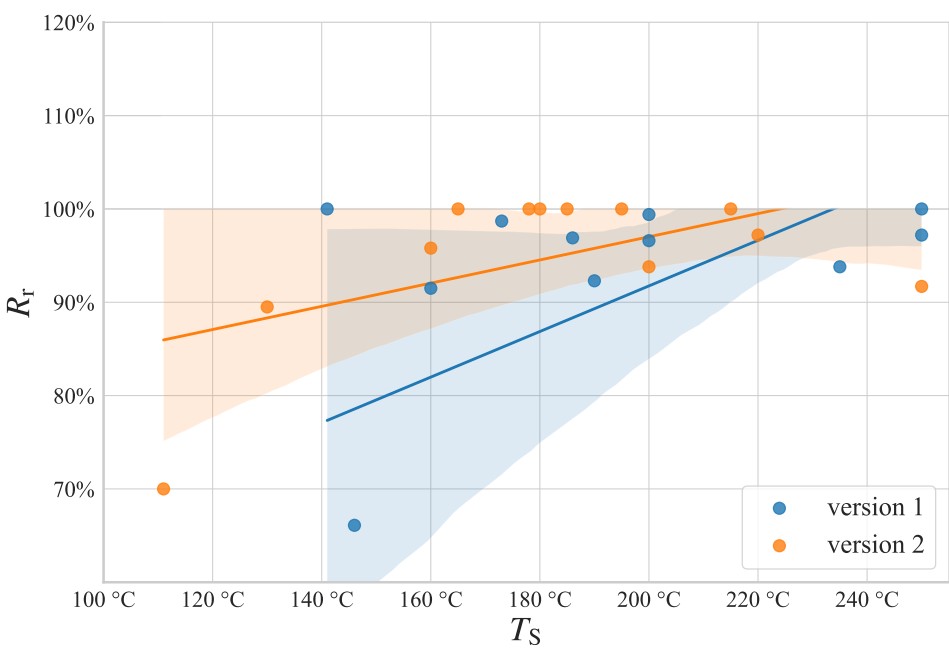

**Figure 10.** The calculated recovery ratio $R_r$ in relation to the measured surface temperature before programming $T_S$ with a linear regression and a 95% confidence interval.

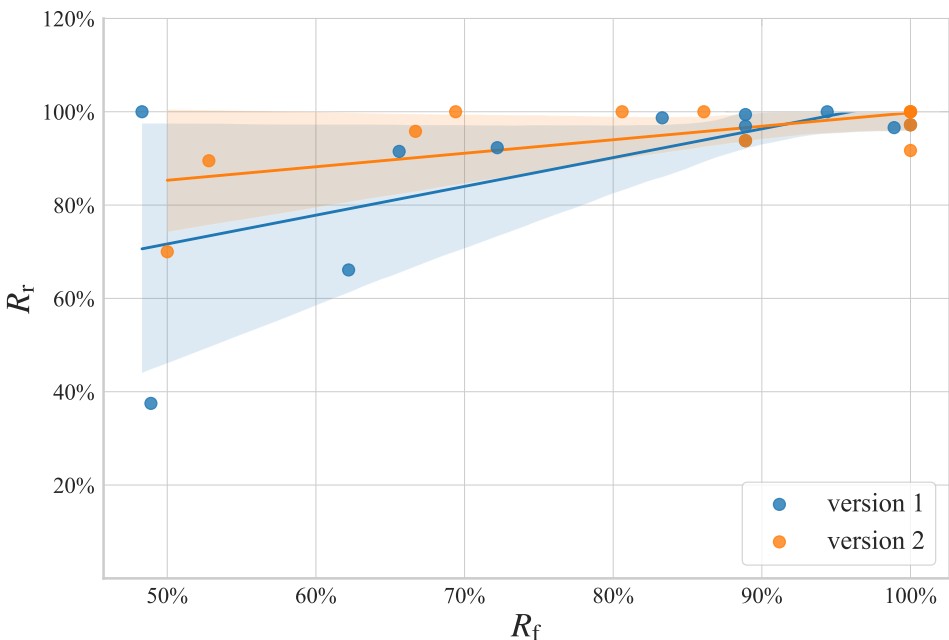

**Figure 11.** The calculated recovery ratio $R_r$ in relation to the calculated fixity ratio with a linear regression and a 95% confidence interval.

### 3.3. Recovery Assessment

After exploring the programming capabilities of the hinges, next, the recovery behavior was investigated. But for these experiments, the results from the previous investigations were used. Obviously, the hinges had to be programmed before they could recover the induced deformation. The temperature $T_{set}$ and the exposure time $t_e$ chosen for this purpose were based on the results of the first experiment. The best programming behavior for the first iteration of the programming device was achieved at $T_{set} = 250\,°C$ and $t_e = 10\,min$. In contrast, a similar fixity ratio was achieved using the second iteration programming device with a shorter exposure time of only $t_e = 5\,min$. Notably, the shorter exposure time

contributed positively to the (energy) efficiency of the whole 4D printing process, which is why the shorter $t_e$ was used for the second version of the programming device.

The hinges were programmed at their respective temperatures and times, always achieving a fixity ratio of close to 100%, with the lowest fixity ratio being 94%.

After exploring the programming of the hinges, we shifted our attention to the recovery process, the main purpose of this experiment. As explained earlier, the hinges were allowed to recover the deformation at different lowered temperatures. Figure 12 shows how the achieved recovery ratio changed with the decreasing temperatures set on the printer's interface for recovery $T_{set(r)}$.

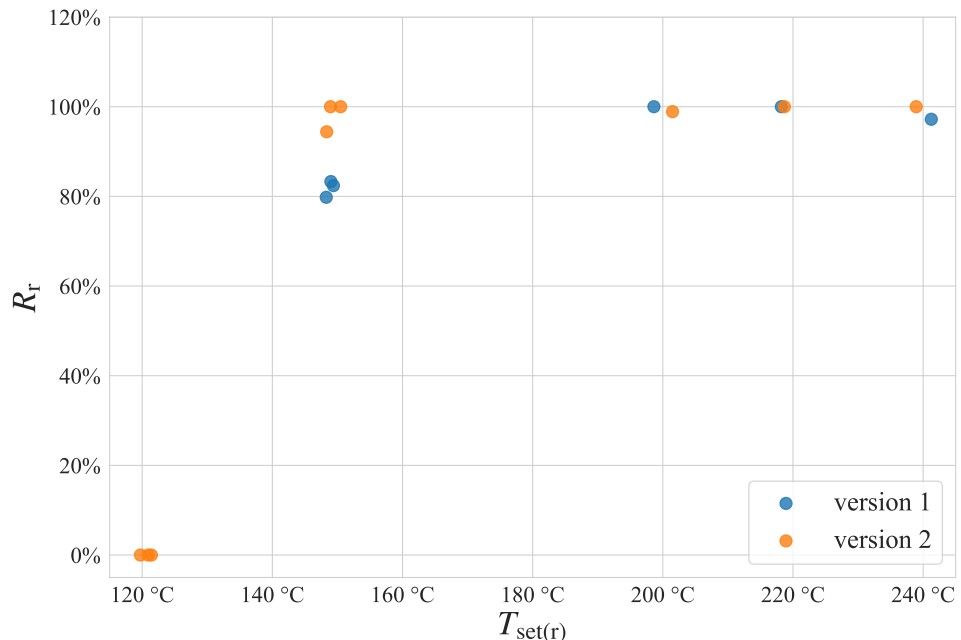

**Figure 12.** The calculated recovery ratio $R_r$ in relation to the input recovery temperature $T_{set(r)}$ with a linear regression and a 95% confidence interval.

Surprisingly, we can see in Figure 12 that the lowered recovery temperature had almost no influence on the recovery ratio of the hinges, with the hinges even achieving full recovery up to 150 °C. However, the 100% recovery was only achieved at 150 °C using the second iteration programming device, thus proving again the value of upgrading the programming device.

As previously noted, the recovery time $t_r$ was documented in these experiments. However, the recovery time was not a primary focus for us. In the context of space structures, it is essential that the recovery did not occur abruptly within an exceedingly brief time frame, but instead over a longer period to minimize vibrations. Nonetheless, this was the case across all observed hinges, with the minimum recovery time being 30 s. Although the hinges exhibited significantly faster recovery at elevated temperatures, the recovery duration was approximately twice as long at the lowest achievable temperature of 150 °C.

We also tested the second iteration of the programming device at $T_{set(r)} = 120$ °C, but no recovery was achieved. The hinges were left inside the printer for longer than 5 min, but it quickly became clear that no recovery could be achieved at this temperature.

### 3.4. Forced Cool-Down Assessment

To enhance the efficiency of programming and recovery for the hinges, we designed forced cool-down experiments. As previously mentioned, the hinges underwent programming and recovery at a lower temperature, where the first experiment did not exhibit an optimal 4D printing performance. The chosen lower temperature was $T_{set} = 175$ °C

with an exposure time of $t_e = 10\,\text{min}$ for the first version of the programming device and $t_e = 5\,\text{min}$ for the second version. The rationale behind opting for a suboptimal programming environment was to observe potential enhancements in performance. This choice was motivated by the fact that in an already ideal process, improvements would be challenging to discern.

The outcomes of this experiment are illustrated in Figure 13. The *x* axis delineates various scenarios, where "r" denotes the regular programming method used earlier, "f" signifies the forced cool-down method involving water, and "v1" and "v2" consistently refer to the use of the first or second iteration programming device, respectively. As a reminder, for each scenario, 7 hinges were tested.

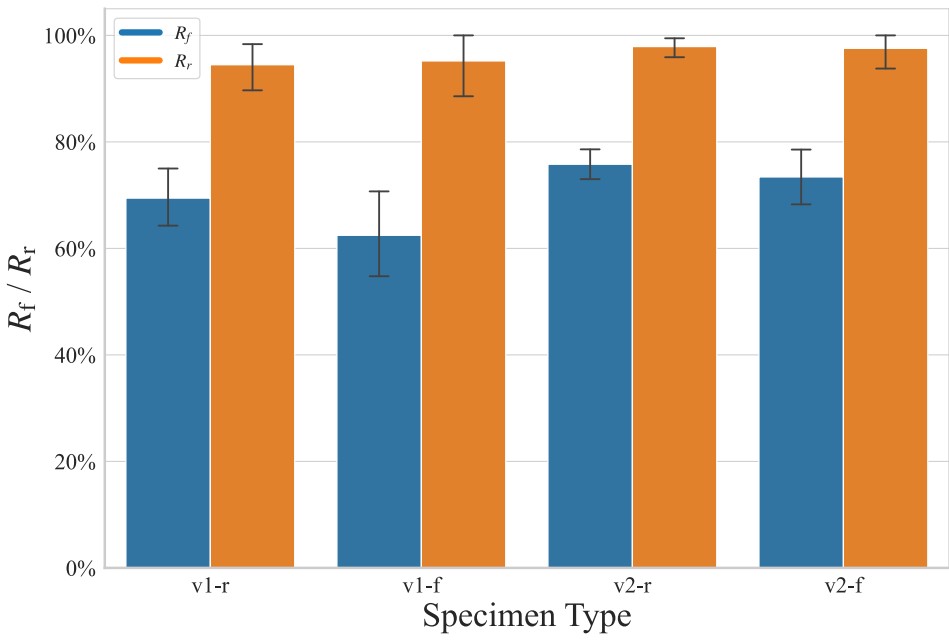

**Figure 13.** The effect of forced cool down on the fixity/recovery ratio $R_f/R_r$ of hinge specimens programmed and recovered at 175 °C with the standard deviation plotted as error bars.

The results are also summarized tabular in Table 2.

**Table 2.** The mean fixity ratio $R_f$ and recovery ratio $R_r$ with their respective standard deviations for the third experiment on the influence of the cool-down force.

| Specimen Type | $R_f$ | SD($R_f$) | $R_r$ | SD($R_r$) |
|---|---|---|---|---|
| M-i 3r | 69.4% | 7.9% | 94.5% | 6.6% |
| M-i 3f | 62.5% | 11.3% | 95.2% | 8.2% |
| M-i 3v2r | 75.8% | 4.2% | 97.9% | 2.8% |
| M-i 3v2f | 73.4% | 8.0% | 97.6% | 5.0% |

## 4. Discussion

### 4.1. Influence of the Programming Device on the 4D Printing Performance

The influence of the programming device on the performance of 4D printed hinges is a critical aspect that emerged from our experiments. A comprehensive analysis revealed distinct differences between the two iterations of the programming device, shedding light on their impact on the programming and recovery processes. Throughout all of the experiments conducted, a consistent observation was the notable inequality in the 4D printing performance between the first and second versions of the programming devices. This difference underscores the significance of the design variations in influencing the overall effectiveness of the process.

An evident trend in the data is the superior performance exhibited by the second version programming device in comparison with its predecessor. This improvement was consistent across all experiments, suggesting a positive correlation between the design enhancements in the second version and the overall efficiency of the 4D printing process.

While the first version facilitated the ease of swapping hinges, a notable drawback was encountered during the programming process. Retracting the movable slider while simultaneously folding the hinges proved to be a laborious and time-consuming task. This operational complexity could have potentially contributed to the observed performance disparities.

A crucial aspect influencing the 4D printing performance was the speed at which the programming device was closed after removal from the 3D printer. With the second iteration programming device, the closing process could be executed slightly faster. This heightened efficiency could account for the improved performance witnessed in experiments conducted with this version.

For the first iteration programming device, thermal expansion of the hinge occasionally resulted in unintended expansion in the x direction. This expansion, occurring during bending, led to a few millimeters of slippage, causing off-center folds. Such deviations underscored the significance of hinge behavior under thermal conditions and emphasized the need for design modifications to mitigate these effects.

The data presented in Figures 10–12 indicate that the programming device not only influences the programming process, but subsequently also significantly impacts the recovery process. The variations observed in recovery times further highlight the intricate interplay between the programming device design and the recovery of the hinges.

This in-depth analysis of the programming devices provides valuable insights into their nuanced influence on 4D printing performance, laying the groundwork for a more comprehensive understanding of the experimental outcomes.

*4.2. Programming*

Our investigation into the programming experiments for 4D printed hinges provides valuable insights into the intricate relationship between temperature variables, programming device design, and the subsequent performance of the SMP hinges.

Examining Figure 7, we observed a noticeable deviation between the measured surface temperature and the temperature set on the printer's interface. This divergence is expected, given the influence of exposure time on achieving the desired temperature during programming. It is crucial to acknowledge that the thermal imaging camera used for surface temperature measurements introduces a margin of error due to variations across the programming device and hinge. Despite this inherent imprecision, the measured temperatures fell within a reasonable range relative to the set temperature on the printer's interface.

Analyzing Figure 8, a clear correlation emerged between the temperature during folding/bending and the quality of bending results. Higher temperatures during bending yielded superior outcomes, aligning with the principles of shape memory effects and supported by the DMTA results gathered by Pigliaru et al. [12] for PEEK (see Figure 14 PEEK_neat).

A notable distinction between the first and second iteration programming device was evident, with the second iteration consistently outperforming the first. The second iteration achieved a 100% programming ratio more frequently and at lower temperatures, underscoring the positive impact of design modifications. In Figure 9, an independent assessment of the programming results, irrespective of the measured surface temperature, vividly showcased the enhanced performance of the second version of the programming device.

The results presented in Figures 10 and 11 highlight the profound influence of programming on hinge recovery. Inadequate programming led to insufficient recovery, emphasizing the critical importance of precision in the programming phase. Conversely, when the programming was optimal, as seen in experiment 2, recovery became almost independent of the recovery temperature. This observation underscored the interdependence

of programming and the subsequent recovery processes in achieving the desired shape memory effects.

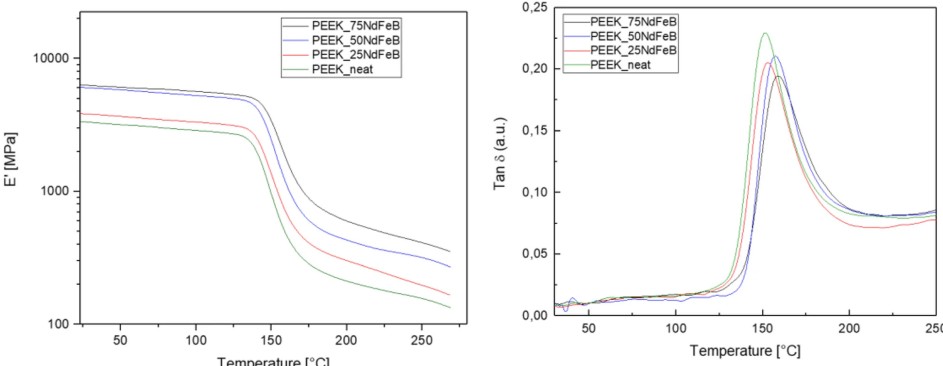

**Figure 14.** "DMTA results: Storage modulus (**left**) Tan tan (**right**) of PEEK_NdFeB filaments" by Pigliaru et al. [12], licensed under https://creativecommons.org/licenses/by/4.0/, (accessed on 13 September 2023) (CC BY 4.0).

*4.3. Recovery*

The recovery experiments conducted offer profound insight into the shape memory capabilities of 4D printed hinges. The results, particularly from the second experiment, reveal noteworthy characteristics that extend the potential utility of these SMP hinges.

In the second experiment, a remarkable observation emerged—the hinges exhibited consistent recovery of almost the full previously induced deformation up to a lower temperature barrier of 150 °C. This effect was notably more pronounced for the second version of the programming device. This finding is intriguing as it surpasses conventional expectations and showcases the remarkable shape memory capabilities of SMP hinges. The observed independence of recovery from the recovery temperature up to the lower barrier of 150 °C is a notable outcome. This characteristic enhances the potential use-cases of these 4D printed hinges, as it suggests a broader operational range for shape memory activation. The increased flexibility in recovery conditions contributes to the adaptability and versatility of these hinges in various applications.

The shape memory capabilities exhibited in experiments 1 and 2 surpassed expectations and expand the perceived boundaries of what was previously imagined possible. This remarkable performance stands in contrast with the sobering DMTA results and anticipated SMEs based on prior research [17]. The experimentally observed recovery at lower temperatures challenges existing assumptions and opens new avenues for exploration in the field of shape memory polymers.

The higher-than-expected shape memory effects observed in our experiments may be attributed to specific design and material properties. The design of the hinges, characterized by their high slenderness, could contribute to the enhanced shape memory effects. Therefore, the heightened slenderness could support the inherent high elasticity of PEEK, allowing for greater deformation and recovery. In addition to slenderness, the material properties, particularly the molecular structure of the specific PEEK variant used in our case, the Evonik Infinam PEEK 9359 F, play a crucial role in the hinge's performance. The unique molecular composition of this PEEK variant likely enhances its shape memory characteristics, contributing to the exceptional recovery observed in our experiments.

The decision to test at 120 °C for the second version of the programming device aligns with the potential use case in space structures. However, the lack of recovery at this temperature is unsurprising given the known glass transition temperature ($T_g$) of PEEK at roughly 150 °C.

The outcomes of the recovery experiment not only validate the exceptional shape memory capabilities of the 4D printed hinges, but also introduce new considerations for their deployment in diverse scenarios.

### 4.4. Forced Cool-Down

While these experiments aimed to expedite the cooling process, the observed outcomes warrant careful consideration. The forced cool-down experiments serve as supplementary investigations, providing additional depth to our understanding of shape memory effects in the context of our experiment setup. These experiments, although not central to the primary objectives, contribute nuanced information about the responsiveness of the 4D printed hinges to rapid cooling conditions.

Unfortunately, the results presented in Figure 13 reveal a close-to-negligible or only a slight negative influence of the forced cool-down method on both the recovery ratio ($R_r$) and folding ratio ($R_f$). Despite the expectation that the forced cool-down method could expedite the cooling process, the hinted worsened performance casts a shadow on its practical benefits. The slight negative impact observed in the 4D printing performance following forced cool-down experiments could be attributed to several factors. Rapid cooling rates may induce thermal stress, potentially leading to incomplete shape fixation. Additionally, variability in cooling conditions introduced by the forced cool-down method might contribute to the observed performance variation.

The observed negative impact prompts consideration for adjustments in the forced cool-down method. Exploring variations in the cooling rate and duration could be avenues for mitigation, allowing for a more refined understanding of the optimal conditions for shape memory activation. These potential refinements not only address the observed limitations, but also open avenues for future experiments aimed at optimizing the forced cool-down process.

Forced cool-down experiments, while revealing a nuanced interplay between cooling dynamics and 4D printing performance, set the stage for further methodological refinement and exploration.

### 4.5. Derived Use-Case Scenarios in Spacecrafts

The application of self-activating hinges in spacecrafts, particularly in the context of deploying structures such as solar arrays, emerges as a promising use case. The unique combination of exceptional mechanical and chemical qualities of PEEK positions it as a viable material for spacecraft applications. Its resilience in extreme environmental conditions, including vacuum and radiation exposure, aligns with the rigorous demands of space missions.

The observed SMEs of the 4D printed hinges exceeded the initial expectations. Achieving a 100% programming and recovery ratio with a recovery temperature of a minimum of 150 °C positions, indicates these hinges are robust components for self-deploying structures in space.The well-programmed hinges cover a full 180° range, making them versatile for a range of applications in spacecrafts. The geometry employed in this study serves as a baseline, acknowledging that adjustments may be necessary depending on the specific use case. A primary envisaged use case involves employing these self-activating hinges for solar arrays on spacecrafts. As solar arrays are exposed to solar radiation after deployment, the hinges autonomously facilitate the opening of the solar arrays, streamlining the deployment process.

While the hinges demonstrate a passive activation temperature of 150 °C, achieving this temperature passively in space environments, especially in Earth orbiting satellites, proves challenging. In Lower Earth Orbit (LEO) or Geostationary Orbit (GEO), temperature ranges may not naturally reach the required 150 °C (refer to [25]). Potential solutions include incorporating PEEK as part of a partial-transition mechanism (PTM) (refer to [19]) in a multi material hinge or utilizing electromagnetic effects for controlled heating.

The exploration of self-activating hinges in spacecrafts unveils exciting possibilities, with PEEK's material properties and the observed shape memory effects presenting a significant advancement in deployable structures for space applications.

## 5. Conclusions

In conclusion, the development and assessment of 4D printed hinges have yielded compelling insights into their programming, recovery behavior, and potential applications in spacecraft structures.

- The two iterations of the programming device played a pivotal role in influencing the 4D printing performance, with the second version showcasing superior results across all experiments.
- The correlation between programming temperature and the quality of bending outcomes underscored the delicate balance in achieving optimal shape memory effects.
- Additionally, the unexpected independence of recovery from the recovery temperature up to 150 °C introduced new considerations for deploying these hinges in diverse scenarios.
- The forced cool-down experiments, while revealing a nuanced interplay between cooling dynamics and 4D printing performance, indicated a potential need for methodological refinement.
- The derived use-case scenarios in spacecrafts highlighted the promising application of self-activating hinges, particularly in deploying structures such as solar arrays.
- In addition to the aforementioned insights, it is noteworthy to acknowledge the instrumental role of the A150 3D printer from Orion Additive Manufacturing in this study. A150 is an innovative tool in the realm of 4D printing, paving the way for more streamlined and efficient experimentation in the field of 4D printing.

**Author Contributions:** Conceptualization, T.R. and C.V.; methodology, T.R.; software, T.R.; validation, T.R. and C.V. data curation, T.R.; writing—original draft preparation, T.R.; writing—review and editing, T.R. and C.V.; visualization, T.R.; supervision, C.V. All of the authors have read and agreed to the published version of the manuscript.

**Funding:** This research received no external funding.

**Data Availability Statement:** The data presented in this study are openly available on GitHub at the following link: https://github.com/richtTim/4D-Printing-of-SMP-Hinges (accessed on 14 November 2023).

**Acknowledgments:** We extend our sincere gratitude to Orion Additive Manufacturing GmbH for generously providing access to their A150 3D printer, offering troubleshooting assistance, and their steadfast support during challenging moments. Their collaboration significantly enriched the depth of our study. We are equally grateful to Evonik Industries AG for supplying the crucial PEEK material, which played a pivotal role in the successful execution of our experiments.

**Conflicts of Interest:** The authors declare no conflicts of interest.

## Abbreviations

The following abbreviations are used in this manuscript:

| | |
|---|---|
| CAD | Computer-aided design |
| CNC | Computerized numerical control |
| DMA | Dynamic mechanical analysis |
| DMTA | Dynamic mechanical thermal analysis |
| FDM | Fused deposition modelling |
| FFF | Fused filament fabrication |
| GEO | Geostationary orbit |
| ISS | International Space Station |
| LEO | Lower Earth orbit |
| PEEK | Polyetheretherketon |
| PLA | Polylactic acid |

| PTM | Partial-transition mechanism |
| SMA | Shape memory alloy |
| SME | Shape memory effect |
| SMP | Shape memory polymer |
| TMA | Thermal mechanical analysis |
| TRH | Thermal radiation heating |

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
