# Peer review of "Development and Assessment of a 4D Printing Technique for Space Applications"

_applsci, doi:10.3390/app14010339_

Round 1

Reviewer 1 Report

Comments and Suggestions for Authors

The authors reported the manuscript – Development and Assessment of a 4D Printing Technique for Space Applications – and developed an efficient device for programming SMP hinges crafted from PEEK polymer, with optimizing the performance for space applications. The presented work and given data are promising and matches with the scope. However, the authors should address the following issues to improve their manuscript. A major revision is required.

1- Add the full name before the abbreviation PEEK in Abstract.

2- Provide all numerical findings in Abstract.

3- What about the application in very low temperatures

4- Give more examples on the types of polymers that can be applied in 4D printing with the related references.

5- In Introduction, the authors should delete the phrases – refer to subsection … – and mention the direct meaning with supportive references.

6- Rewrite the Section – 2.1. Materials – providing only the type, supplier and some properties of the materials used. The mentioned discussion should be added to the Results section.

7- Highlight the novelty of the research.

8- Remove Figure 1 to the Discussion part.

9- The Conclusion is very long. Modify to give the main findings without details.

Author Response

  1. Summary

  1. Point-by-point response to Comments and Suggestions for Authors

Comment 1: Add the full name before the abbreviation PEEK in Abstract.

Response 1: Thank you for pointing this out. We agree with this comment. Therefore, we have added the name Polyetheretherketon before the abbreviation in the manuscript.

Comment 2: Provide all numerical findings in Abstract.

Response 2: We appreciate the feedback and acknowledge the importance of clarity in presenting numerical findings. We have included the main numerical findings in the abstract. Additionally, we have incorporated a statement that directs readers to the specific sections in the paper where more detailed information on these numerical results can be found.

Comment 3: What about the application in very low temperatures.

Response 3: Thank you for pointing out the application in very low temperatures. As detailed in the Discussion section, it is important to note that the self-activating behavior of the hinges is observed at a temperature of 150°C. However, the PEEK material, despite its specific behavior at higher temperatures, remains a versatile elastic polymer. Furthermore, there is potential for combining PEEK with other materials that exhibit desirable characteristics at lower temperatures, broadening the scope of potential applications.

Comment 4: Give more examples on the types of polymers that can be applied in 4D printing with the related references.

Response 4: In response to this suggestion, we have incorporated a section within the manuscript that delves into various types of polymers suitable for Shape Memory Applications in 4D printing. This section not only provides examples of other polymers which can be used but also includes references to relevant literature, offering readers a broader understanding of the diverse range of polymers applicable in our discussed context.

Comment 5: In Introduction, the authors should delete the phrases – refer to subsection … – and mention the direct meaning with supportive references.

Response 5: Thank you for pointing this out. I/We agree with this comment. Therefore, we have deleted theses phrases from the introduction.

Comment 6: Rewrite the Section – 2.1. Materials – providing only the type, supplier and some properties of the materials used. The mentioned discussion should be added to the Results section.

Comment 7: Remove Figure 1 to the Discussion part.

Response 6/7: Thank you for these suggestions. We have carefully considered the feedback and have made the decision to relocate Figure 1 along with the pertinent text snippet from Section 2.1 (Materials) to the Discussion section. However, after careful consideration, we have opted to retain the remaining content in Section 2.1, as we believe that removing it entirely might disrupt the coherent flow of the paper.

Comment 8: Highlight the novelty of the research.

Response 8: Thank you for pointing this out. I/We agree with this comment. In response, we have incorporated a dedicated paragraph in the introduction that explicitly emphasizes the unique aspects and contributions of our work.

Comment 9: The Conclusion is very long. Modify to give the main findings without details.

Response 1: We appreciate you bringing this to our attention. We agree and modified the conclusion to only include the main findings, without as much detail.

Reviewer 2 Report

Comments and Suggestions for Authors

-        Lines 16-34 require citations.

-        More information about SMP material is required, as well as justification and reasons for using PEEK material.

-        Is there any effect between the changing geometry of hinges in geometry a and b in figure 4?

-        Printing parameters (speed, nozzle temperature, and bed temperature) should be entered.

-        What is the 3D printing infill pattern and programming device infill density?

-        The total thickness is 0.4, what is the thickness of the top and button layers, and what is the layer height?

-        Based on figures 6 a, and c, you utilized wooden and plastic parts, what is the reason, and where did you use part b in Figure 4, could you add a figure?

Author Response

  1. Summary

  1. Point-by-point response to Comments and Suggestions for Authors

Comment 1: Lines 16-34 require citations.

Response 1: Thank you for pointing this out. We agree with this comment and have added the relevant literature to the introduction.

Comment 2: More information about SMP material is required, as well as justification and reasons for using PEEK material.

Response 2: We acknowledge the reviewer's feedback and have addressed the need for more information on SMP materials. The revised manuscript includes a paragraph detailing heat induced SMP materials, their characterization, and examples. Additionally, we've added a separate paragraph justifying the selection of PEEK.

Comment 3: Is there any effect between the changing geometry of hinges in geometry a and b in figure 4?

Response 3: Thank you for bringing attention to this aspect. We confirm that the changing geometry of hinges in Figures 4a and 4b has no significant effect on our results. The holes were intentionally positioned far from the bending axis. To address this, we've added a brief paragraph in the manuscript for clarification.

Comment 4: Printing parameters (speed, nozzle temperature, and bed temperature) should be entered.

Comment 5: What is the 3D printing infill pattern and programming device infill density?

Comment 6: The total thickness is 0.4, what is the thickness of the top and button layers, and what is the layer height?

Response 4/5/6: We appreciate the concern regarding the printing parameters. In response to Comment 4, we acknowledge the importance of providing printing parameters, including speed, nozzle temperature, and bed temperature. However, as outlined in Section 2.3 of the manuscript, we currently find ourselves bound by non-disclosure agreements with Orion Additive Manufacturing GmbH, prohibiting the specific disclosure of 3D printing parameters. To maintain transparency, we have included information about the infill pattern and density in Section 2.3, in agreement with Orion.

Comment 1: Based on figures 6 a, and c, you utilized wooden and plastic parts, what is the reason, and where did you use part b in Figure 4, could you add a figure?

Response 1: Thank you for pointing this out. To clarify, the brown parts in Figures 6a and 6c are 3D-printed PEEK components, not wooden parts. We acknowledge the oversight and have amended the figure labels to clarify that the components are made from PEEK.

Round 2

Reviewer 1 Report

Comments and Suggestions for Authors

The authors have addressed all comments according to the provided suggestions. The manuscript can be accepted in the current version. 

Reviewer 2 Report

Comments and Suggestions for Authors

This material is common, so it is preferable to add more information for printing parameter